# Nurses’ Knowledge, Attitudes, and Practices in Pressure Injury Prevention: A Systematic Review and Meta-Analysis

**DOI:** 10.3390/healthcare13111220

**Published:** 2025-05-22

**Authors:** Mousa Yahya Asiri, Omar Ghazi Baker, Homoud Ibrahim Alanazi, Badr Ayed Alenazy, Sahar Abdulkareem Alghareeb, Hani Mohammed Alghamdi, Saeed Bushran Alamri, Turki Almutairi, Hussien Mohammed Alshumrani, Muhanna Alnassar

**Affiliations:** 1College of Nursing, King Saud University, Riyadh 12372, Saudi Arabia; obaker@ksu.edu.sa (O.G.B.); mmalnassar@ksu.edu.sa (M.A.); 2Nursing Department, King Saud University Medical City, Riyadh 12372, Saudi Arabia; hoalanazi@ksu.edu.sa; 3Northern Borders Health Cluster, Arar 73211, Saudi Arabia; balenazy@moh.gov.sa; 4College of Nursing, Imam Abdulrahman Bin Faisal University, Dammam 31451, Saudi Arabia; saalghareb@iau.edu.sa; 5Erada Complex and Mental Health, Third Health Cluster, Riyadh 12571, Saudi Arabia; halghamdi23@moh.gov.sa; 6Majaredah General Hospital, Aseer Cluster, Al-Majardah 63931, Saudi Arabia; sabualamri@moh.gov.sa; 7Department of Medical Surgical, College of Nursing, Qassim University, Buraydah 52555, Saudi Arabia; tu.almutairi@qu.edu.sa; 8Ministry of Health Office in Bisha Provenance, Bisha 67755, Saudi Arabia; halshamrani@moh.gov.sa

**Keywords:** attitude, knowledge, meta-analysis, nurse, practice, pressure injury, prevention, systematic review

## Abstract

*Background:* Various methods for preventing pressure injury have been developed across the globe, particularly in Saudi Arabia. Current available research has investigated the knowledge, attitudes, and practices regarding pressure injury prevention. However, no systematic review and meta-analysis have yet examined the associations among knowledge, attitudes, and practices regarding pressure injury prevention based on the perspectives of registered nurses. *Objective:* This study examines and summarizes the reported relationships among knowledge, attitudes, and practices regarding pressure injury prevention on the basis of the perceptions of registered nurses. *Methods:* The CINAHL, ProQuest, PubMed, ScienceDirect, and Web of Science databases were searched for quantitative evidence published in English between 2019 and 2025. The systematic review and meta-analysis adhered to the Preferred Reporting Items for Systematic Reviews and Meta-Analyses (2021) guidelines. *Results:* Out of the 1986 records that were initially examined, a total of 10 quantitative, cross-sectional, and correlational studies were included in the final systematic review and meta-analysis. In the context of the meta-analysis, 10 studies were included for the association between knowledge and attitudes, whereas only 3 studies were available for the association between knowledge and practice, and similarly, only 3 studies addressed the association between attitudes and practice of pressure injury prevention. Collectively, 2457 registered nurses were involved in these studies, mostly working in intensive care units. The studies were conducted in various countries across Asia and the Middle East, mostly in Turkiye, within the last five years. The registered nurses in the 10 studies reported associations among knowledge, attitudes, and practices toward pressure injury prevention that ranged from insignificant to weak, indirect, and strong direct. *Conclusions:* Relationships among knowledge, attitudes, and practices toward pressure injury prevention are both positive and negative from the global perspective and are shaped by various confounding and mediating factors, including socio-demographic, nursing-related, and hospital-related factors. Improving the knowledge base of registered nurses and promoting a favorable attitude toward pressure ulcer prevention would provide healthcare organizations with the potential to enhance the already commendable levels of practice and prevention noted in this review.

## 1. Introduction

A pressure injury, commonly referred to as a pressure ulcer or bedsore, is localized damage to the skin and underlying tissue, occurring because of sustained pressure, particularly in areas over bony prominences [1]. Pressure injuries are prevalent in healthcare settings, and international incidence rates range from 4.7% to 31% [2]. Research has underscored the considerable economic implications of these injuries for patients and healthcare systems. For example, estimates suggest that the total cost associated with hospital-acquired pressure injuries reached over USD 26.8 billion in 2019 in the United States [3], which now allocates approximately USD 11 billion annually for the treatment of these injuries [3,4]. Consequently, pressure injuries represent a critical issue within healthcare environments, given their serious implications for patient morbidity, mortality, and the financial strain they impose on healthcare systems [5]. Globally, these injuries constitute a major health concern because of their complications, ranging from pain to infection and potentially fatal outcomes [6].

Pressure injuries are particularly prevalent among hospitalized patients with restricted mobility. Consequently, the role of nurses in the implementation of pressure injury prevention strategies is essential, making their knowledge, attitudes, and practices (KAP) critical to improving patient outcomes [7,8]. In Saudi Arabia, the increasingly aging population and rise in chronic health conditions underscore the critical need for effective strategies to prevent pressure injuries, which continue to pose considerable challenges within the country’s healthcare environments [9]. Thus, the KAP of nurses regarding adherence to pressure injury prevention protocols is vital. However, a comprehensive assessment is needed to determine their effectiveness [9]. The absence of national prevalence data represents a shortfall in the application of established prevention strategies, and research regarding how nurses’ KAP impacts the prevention of pressure ulcers and injuries in inpatient settings across the country is lacking [10]. Investigating this area will help identify educational gaps, develop innovative strategies, and ultimately improve the quality of care on a national scale [11]. In this context, factors such as cultural influences, resource availability, and institutional policies may considerably affect the KAP framework [12].

The literature indicates that nurses typically have a foundational understanding of pressure injury prevention strategies. However, the application of this knowledge in clinical settings is inconsistent [13]. Various factors, such as workload, staffing ratios, training opportunities, and institutional backing, considerably impact adherence to pressure injury prevention [14]. Research has demonstrated a positive relationship between extensive education on pressure injury prevention and improved patient outcomes [15]. However, despite the presence of clinical guidelines, investigations have uncovered a disparity between theoretical knowledge and practical application at the bedside. This discrepancy underscores the necessity for further research into nursing KAP to pinpoint obstacles and enablers that influence the effective implementation of pressure injury prevention strategies [16].

Nurses’ understanding, perceptions, and actions concerning pressure injury prevention are vital elements that affect patient outcomes within healthcare environments. Nurses who maintain positive attitudes toward safety are associated with a decrease in the occurrence of pressure injuries [17]. One systematic review revealed that four out of five studies showed a considerable link between favorable safety attitudes among nurses and lower pressure injury rates in acute care settings [18]. This finding indicates that improving nurses’ safety attitudes can represent a crucial approach for reducing the incidence of pressure injuries. Furthermore, the creation of educational initiatives designed to enhance nurses’ knowledge and competencies in pressure injury prevention is imperative. One study conducted in nursing homes employed the practice change wheel framework to pinpoint the essential factors affecting pressure injury prevention and highlighted the necessity of improved psychological and physical capabilities among nursing personnel [19]. Another investigation demonstrated that structured educational programs markedly enhanced nurses’ performance and patient outcomes concerning pressure injuries related to medical devices [20]. These results emphasize the importance of ongoing education and training in cultivating a well-informed nursing workforce capable of executing effective pressure injury prevention measures.

Despite the acknowledgment of the critical role of knowledge and attitudes, numerous studies indicate a troubling gap between theoretical understanding and practical implementation among nurses. A cross-sectional study in Ethiopia revealed that although nurses displayed a commendable level of knowledge regarding pressure injury prevention, their actual practices were substandard [21]. This inconsistency is reflected in the results of various studies [16,18,19,20], suggesting that effective practice can be impeded by other factors, such as inadequate training, resource limitations, and negative attitudes, even when nurses have sufficient knowledge. In addition to educational interventions, policies that address systemic barriers are essential for enhancing adherence to pressure injury prevention practices. Optimal nurse–patient ratios and continuous training are critical factors that positively affect nursing practices [22]. One study revealed that nurses who participated in formal training on pressure injury prevention exhibited considerably improved knowledge and practices compared with their counterparts who did not receive similar training [23]. This finding underscores the importance of organizational support and structured training programs in cultivating a culture of safety and competence in managing pressure injuries [19].

Enhancing nurses’ KAP regarding pressure injury prevention necessitates a comprehensive approach that encompasses educational initiatives, the removal of systemic barriers, and the promotion of positive safety attitudes [8]. Despite extensive research on pressure injury prevention, a notable gap persists in understanding how nurses’ KAP collectively impact adherence to prevention strategies [24]. Many studies tend to concentrate on isolated aspects, such as knowledge or practice, without integrating these components to evaluate their cumulative effects on adherence [17]. This fragmented perspective hinders the ability of nurses to draw holistic conclusions regarding barriers and facilitators that influence adherence [25]. Addressing these deficiencies is crucial for developing effective and evidence-based interventions that enhance adherence to pressure injury prevention protocols for nursing personnel [26].

Pressure injury prevention within nursing practice encompasses a complex interplay of KAP implementation by nurses. A substantial amount of research underscores the pivotal influence of nurses’ knowledge on their attitudes and practices related to adherence to pressure injury prevention protocols [27]. For example, research conducted by Zhang et al. [28] revealed a positive relationship between nurses’ KAP in preventing pressure injuries associated with medical devices, indicating that an increase in knowledge correlates with improved attitudes and practices. Awoke et al. [29] discovered that nurses lacking adequate knowledge are considerably less inclined to adopt effective pressure injury prevention measures than those with adequate training, thereby highlighting the importance of KAP as essential components of proficient nursing care. However, a conspicuous gap is evident in the literature regarding nurses’ adherence to pressure injury prevention strategies, particularly in relation to the KAP factors that influence this adherence or its absence. Some studies [27,28,29] have addressed the various dimensions of pressure injury prevention, but comprehensive assessments that connect these dimensions to the adherence practices of nursing staff are lacking. This gap emphasizes the urgent need for targeted research to determine how KAP affects adherence and identify effective strategies for enhancing compliance with evidence-based prevention measures [30].

Pressure injury prevention initiatives are evolving globally and specifically within Gulf countries, including Saudi Arabia. The objective of this systematic review and meta-analysis is to investigate the current state of research concerning KAP related to pressure injury prevention to shed light on the development of KAP principles. Given the disparities in existing research, this systematic review and meta-analysis aims to assess the levels of KAP among nurses and the existing associations among these concepts regarding pressure injury prevention from the perspective of the registered nurses.

## 2. Materials and Methods

This systematic review and meta-analysis examined the literature concerning registered nurses’ KAP regarding the prevention of pressure injury. The protocol of this systematic review and meta-analysis was registered with the International Platform of Registered Systematic Review and Meta-Analysis Protocols (INPLASY) on 27 April 2025, with INPLASY registration number INPLASY202540098 and the following DOI: 10.37766/inplasy2025.4.0098.

A systematic review and meta-analysis methodology was employed, which involved a statistical analysis of the results from selected studies that were evaluated according to their quality [31]. As noted by Ahn and Kang [31], systematic review and meta-analysis serves as a statistical technique for synthesizing and interpreting findings from multiple comparable studies.

This systematic review and meta-analysis integrated existing knowledge, summarized findings, and investigated potential gaps concerning registered nurses’ perceptions of their KAP related to pressure injury prevention. The study adhered to the Preferred Reporting Items for Systematic Reviews and Meta-Analyses (PRISMA) guidelines, as outlined by Page et al. [32]. In accordance with PRISMA, this review employed rigorous and systematic approaches for identifying, selecting, evaluating, and synthesizing the included studies to directly address the primary research question: *What are the reported relationships of knowledge, attitudes, and practices regarding pressure injury prevention as perceived by registered nurses?*

### 2.1. Eligibility Criteria

Studies published in a peer-reviewed journal, written in English, and released within the last five years, specifically from 2019 to 2024, were included. This systematic review and meta-analysis, which focuses on pressure injury prevention, appropriately establishes a five-year timeframe to ensure the inclusion of adequate research evidence that is both high-quality and current within the last five years [33]. The focus of these studies was registered nurses’ KAP regarding the prevention of pressure injuries. After a systematic search, quantitative studies were selected for inclusion, which were meta-analyses. Meanwhile, studies not published in English, commentaries, discussion papers, dissertations, narrative reviews, opinion pieces, editorials, secondary analyses of existing data, qualitative studies, and any research that did not address nurses’ perceptions of their KAP related to pressure injury prevention were excluded.

### 2.2. Search Strategy

This review was carried out in January 2025. The researchers collaborated with a university librarian to formulate key search terms for the literature search. The review examined five databases (CINAHL, ProQuest, PubMed, ScienceDirect, and Web of Science), focusing on studies published from 2019 to 2025. As noted by Dialog Solutions [34], a recommended approach for conducting literature searches involves utilizing a minimum of three databases, and PubMed is one of the most frequently used. The search strategy employed a combination of Boolean operators and medical subject headings to identify relevant literature. The terms utilized included “association” OR “relationship” OR “correlation” AND “nurse” OR “registered nurse” OR “nursing staff” OR “nursing personnel” OR “nursing professional” AND “perception” OR “report”, AND “attitude” AND “knowledge” AND “practice” AND “prevention” AND “bedsore” OR “pressure injury” OR “pressure ulcer”. The terms “pressure injury” and “pressure ulcer” are used interchangeably in this review. The Google Scholar search engine, along with the reference lists of the studies included in the review, was utilized to identify further sources. Horsley et al. [35] suggested that review authors examine the reference lists of the studies they have included to enhance their search efforts, particularly when it proves difficult to identify all pertinent studies through manual and database searches.

### 2.3. Data Selection Process

In accordance with the guidelines established by Cochrane [36], two researchers conducted independent evaluations of studies at various stages, specifically during the assessment of titles and abstracts and during the full-text screening process. The evaluation was facilitated through the use of Covidence Systematic Review Software [37] (https://www.covidence.org/), and any disagreement was addressed by a third researcher. Covidence software, which was employed to eliminate duplicate entries, was supplemented by manual checks conducted by the researchers when duplicates were identified. When uncertainty regarding the inclusion of a particular article was found, an inclusive approach to decision making was adopted, fostering discussion. A consensus would be reached before the primary researcher made a final determination. All excluded studies were accompanied by documented reasons for exclusion [32,38], as illustrated in Figure 1.

The PRISMA flow chart illustrates the methodology employed in the search, screening, and selection of studies (see Figure 1). A total of 1986 titles and abstracts from various databases were imported into Covidence for the screening process. After the removal of 231 duplicate entries, 1755 records were subjected to screening. The titles and abstracts were evaluated for their relevance. A total of 52 studies were finally selected for further examination, of which 42 studies were excluded for specified reasons (Figure 1). Ultimately, 10 studies were incorporated into the final systematic review and meta-analysis. The data extraction process encompassed several characteristics of the included articles, such as the author, publication year, country of origin, study objectives, design, key findings related to nurses’ perceptions of KAP concerning pressure injury prevention, limitations, and the Joanna Briggs Institute (JBI) appraisal score [39].

### 2.4. Quality Assessment

Ten studies were ultimately incorporated into this systematic and meta-analytic review, following the quality assessment guidelines established by the JBI [39] for full-text studies. As noted by Tawfik et al. [38], data evaluation involves two or three independent researchers, and a set of articles distinct from those previously analyzed is assigned to each researcher. The selected studies underwent independent and rigorous appraisal by two researchers utilizing the JBI [39] critical appraisal tools (Table 1). Based on the JBI critical appraisal tool for cross-sectional studies, a total of eight questions were employed to appraise each article, with four possible responses for each question: Y (Yes), N (No), U (Unclear), and NA (Not Applicable). The overall appraisal outcome was either to include or exclude the study from the final systematic review and meta-analysis. An article received an excellent appraisal if it achieved a quality assessment of 75% or higher.

### 2.5. Statistical Analyses

The statistical analyses were executed using Comprehensive Meta-Analysis software version 4.0 [49]. In this meta-analytic review, the results were assessed with respect to the odds ratio (OR) and *p*-values. Additionally, cumulative analyses, meta-regression, sensitivity analyses, and subgroup evaluations were effectively used in determining the consistency of the findings, exploring the influence of potential confounding variables on the outcomes of the study, and identifying the most important predictors [38]. Consequently, this methodology was instrumental in investigating nurses’ perceptions regarding their KAP related to the prevention of pressure injury.

Fixed-effect models were employed in statistical analyses for determining the specific OR and the 95% confidence interval (CI) for the dependent variable, which pertained to the prevention of pressure injury as perceived by registered nurses. Furthermore, publication bias was assessed using funnel plots and Egger tests, given that the review encompassed fewer than ten studies addressing the same outcome related to pressure injury prevention. A *p*-value of less than 0.10 from the Egger test was interpreted as indicating potential bias [49,50]. In addition, under fixed-effect model, it is posited that true effects do not vary. Hence, all indices of heterogeneity, including I-squared, are assumed to be zero [50].

## 3. Results

From the initial search outcome of 1986 records, a total of 10 studies were included in the final systematic review. Similarly, within the scope of the meta-analysis, 10 studies were included to explore the relationship between knowledge and attitudes [30,40,41,42,43,44,45,46,47,48]. However, only three studies were eligible to be included that investigated the relationship between knowledge and practice [40,44,46]. Likewise, only three studies focused on the relationship between attitudes and the practice of pressure injury prevention [40,44,46]. Overall, a total of 2457 registered nurses were involved in these quantitative, cross-sectional, and correlational studies; the majority worked in critical care units. The studies were conducted in various countries across different continents, including Africa in Burundi [44]; Europe in Slovakia [45]; and Asia in the Middle East, such as Iran [30,46] and Saudi Arabia [42,43], but mostly in Türkiye [40,41,47,48], within the last five years (i.e., 2019–2024).

To measure nurses’ knowledge about pressure injury prevention, the Pressure Ulcer Prevention Knowledge Assessment Instrument (PUPKAI-T) was utilized in two studies [40,47]. The Pressure Ulcer Knowledge Assessment Tool (PUKAT) 2.0 was employed in three studies [41,42,45]. In addition, the Nurses’ Knowledge of Pressure Ulcer Prevention & Management Questionnaire was applied in one study [43]. The Pieker Pressure Ulcer Knowledge Test (PPUKT) was used in two studies [30,46]. Furthermore, a self-report questionnaire assessing knowledge on pressure injury prevention, based on two established clinical practice guidelines—the Pan Pacific Guideline for the Prevention and Management of Pressure Injury (2012) from the Australian Wound Management Association and the Association for the Advancement of Wound Care (AAWC)—was utilized in one study [44]. Lastly, a tool for pressure ulcer information was implemented in one study [48]. All tools utilized to evaluate nurses’ knowledge regarding pressure injury prevention were reportedly validated.

To assess nurses’ attitudes toward pressure injury prevention, the following tools were used and validated, including the Attitude Toward Pressure Ulcer Prevention Instrument (APuP) [30,40,41,42,45,46,47,48], the Nurses’ Attitude of Pressure Ulcer Prevention Questionnaire [43], and the attitude toward pressure injury prevention, based on two established clinical practice guidelines by the Pan Pacific Guideline for the Prevention and Management of Pressure Injury (2012) from the Australian Wound Management Association and the Association for the Advancement of Wound Care (AAWC) [44]. The tools used to quantitatively assess nurses’ practices on pressure injury prevention were only employed in three studies, including the Nurses’ Practice of Pressure Ulcer Prevention Questionnaire [43], the Practice of Pressure Ulcer Prevention [46], and the practice on pressure injury prevention, based on two established clinical practice guidelines by the Pan Pacific Guideline for the Prevention and Management of Pressure Injury (2012) from the Australian Wound Management Association and the Association for the Advancement of Wound Care (AAWC) [44].

As shown in Table 2, Zencir et al. [40] revealed that the surgical nurses’ knowledge concerning pressure injury prevention was insufficient in Turkiye. Although the nurses exhibited a favorable attitude toward pressure injury prevention, this positive outlook did not reach a satisfactory level. Furthermore, the level of knowledge possessed by the nurses served as an important predictor of their positive attitudes.

In another study in Turkiye, Şahan and Güler [41] reported that the mean total score for nurses on PUKAT 2.0 was 9.40 and the standard deviation was 2.47, indicating a correctness rate of 46.72%. In contrast, the mean total score for nurses on the APuP instrument was 32.39, which had a standard deviation of 2.752. A significant positive correlation was observed between the total knowledge and total attitude scores, with a correlation coefficient (ρ) of 0.761 and a *p*-value of 0.007. Furthermore, the linear regression analysis results revealed that gender, educational background, years of experience, and unit of employment are important predictors of PUKAT 2.0 and APuP scores.

In Saudi Arabia, Alshahrani et al. [42] reported that the mean pre-intervention scores reflecting nurses’ knowledge and attitudes regarding pressure injury prevention were 43.22% and 74.77%, respectively. Subsequent to the educational intervention, these scores exhibited a considerable increase, reaching 51.22% for knowledge and 79.02% for attitudes. An enhanced understanding of pressure injury prevention was positively correlated with favorable attitudes toward preventive practices. Other factors, such as age, clinical nursing experience, and exposure to intensive care settings, were significantly correlated with knowledge in this area. Furthermore, possessing a bachelor’s degree or higher was associated with improved knowledge and attitudes concerning pressure injury prevention.

In another study from Saudi Arabia, Zabidi et al. [43] revealed that only a small proportion of participants displayed an adequate understanding of pressure ulcer prevention, and merely one-third showed a favorable attitude toward preventive measures. In contrast, the majority of participants exhibited satisfactory levels of practice and adherence to protocols related to the prevention and management of pressure ulcers. The findings indicated a notable correlation between attitudes toward pressure ulcer prevention and marital status and the presence of training in this area (*p* < 0.05). Divorced nurses and those who had received formal training in pressure ulcer prevention exhibited a positive attitude. Furthermore, the analysis revealed considerable differences in the variables of gender, marital status, educational attainment, training on pressure ulcer prevention, and area of practice, as measured using the pressure ulcer practice scale (*p* < 0.05).

Ghazanfari et al. [30] reported that the mean scores reflecting the KAP of intensive care unit (ICU) nurses in Iran concerning pressure injury prevention were 70.57 (SD = 13.51), 52.82 (SD = 6.16), and 22.44 (SD = 5.20), respectively. A significant positive correlation was identified between the attitudes and practices of the nurses (r = 0.232, *p* = 0.002), whereas a negative correlation was observed between their knowledge and attitudes (r = −0.156, *p* = 0.035) regarding pressure injury prevention. Furthermore, a positive correlation was established between the duration of the nurses’ experience in the ICU and their knowledge of pressure injury prevention (r = 0.159, *p* = 0.032).

In Burundi, Africa, Niyongabo et al. [44] indicated notably low scores for knowledge and practice among nurses, and the participants achieved less than 50% on the six items assessing knowledge and the six items evaluating practice. In contrast, the scores reflecting attitudes were higher, exceeding 65% on the five items designed to measure this aspect. A significant negative correlation was identified between the knowledge and attitude scores of the nurses (r = −0.479, *p* = 0.015). Furthermore, the level of education was inversely related to the knowledge and practice scores concerning the prevention and treatment of pressure ulcers. Notably, a high attitude score did not correspond to an increased practice score, probably because of the low knowledge scores observed (below 50% on the knowledge items).

In Slovakia, Halász et al. [45] indicated that nurses exhibited inadequate knowledge (45.5%) and attitudes (67.9%) regarding pressure injury prevention. A significant positive correlation was identified between their knowledge and attitudes (*p* = 0.300; *p* < 0.001). Furthermore, the level of education (*p* = 0.031) and the specific work department (*p* = 0.048) were associated with notable differences in the nurses’ knowledge levels.

In another study from Iran [46], the mean scores reflecting the KAP of nurses regarding pressure injury prevention were 63.47 ± 10.31, 39.10 ± 40.22, and 32.03 ± 6.17, respectively. A significant positive correlation was identified among the three variables. The results indicated that for each additional year of experience in the ICU, nurses’ knowledge increased by 0.051 units. Furthermore, female nurses demonstrated higher levels of knowledge and attitude than their male counterparts, with differences of 3.132 and 1.65 units, respectively. Additionally, the attitude of nurses improved by 0.43 units for every extra hour worked per week. Nurses in the general ICU exhibited superior attitude and practice scores compared to other nursing staff, with increases of 2.144 and 2.574 units, respectively. Lastly, the practice of nurses increased by 0.162 units for each additional year of age.

In their study in Turkey, Aydoğan and Çalışkan [47] reported a mean knowledge score of 11.54 and a standard deviation of 2.91, and the mean attitude score was 42.96 with a standard deviation of 4.06. The primary barriers identified in pressure injury prevention included inadequate staffing levels, as reported by 85.6% of respondents, and a lack of pressure redistribution materials and equipment, as noted by 82.6%. A regression analysis of the attitude scores indicated that several factors influenced nurses’ attitudes toward pressure injury prevention: self-sufficiency in conducting pressure ulcer risk assessments (B = 0.28), a desire to acquire further knowledge on pressure injury prevention (B = −0.15), gender (B = −0.15), and the knowledge score itself (B = 0.14).

Another study conducted in Turkey by Yilmazer et al. [48] reported that nearly half of the nurses (48.1%) working in ICUs demonstrated inadequate knowledge regarding pressure injury prevention. Furthermore, fewer than one-quarter (21%) of the nurses achieved attitude scores of 75% or higher. A significant negative correlation was identified between the knowledge levels and attitudes of nurses concerning pressure injury prevention (*p* < 0.05). This correlation indicated that as nurses’ knowledge about pressure injury prevention increases, their positive attitudes tend to diminish.

The meta-analysis findings and corresponding forest plot presented in Figure 2, Figure 3 and Figure 4 illustrate the perceptions of nurses concerning their KAP in relation to the prevention of pressure injuries. In this meta-analysis, particularly in the utilization of research tools (e.g., PUKAT 2.0, APuP, and PUPKAI-T), the Q-value (1.016) is less than two degrees of freedom, and the amount of between-study variance in the observed effects is actually less than would be expected to be seen based on sampling error alone [49,51]. There is no evidence that the effect size varies across 10 studies. Notably, the nurses’ KAP was linked to an increased likelihood of preventing pressure injuries.

All 10 studies included in the final review showed significant results for the correlation between knowledge and attitudes in pressure injury prevention, as demonstrated by the fixed-effect models shown in Figure 2 (correlation value = 0.389; CI = 0.354–0.422; *p* < 0.001).

Regarding the correlation between knowledge and practice regarding preventing pressure injuries, only three studies reported correlation values [40,43,44]. These three studies demonstrated significant results for the correlation between knowledge and practice in pressure injury prevention according to fixed-effect models, as illustrated in Figure 3 (correlation value = 0.403; CI = 0.332 to 0.469; *p* < 0.001).

Moreover, only three studies reported correlation values for the correlation between attitudes and practice in preventing pressure injuries [40,43,44]. These three studies showed significant results for the correlation between knowledge and practice in pressure injury prevention, as demonstrated by fixed-effect models illustrated in Figure 4 (correlation value = 0.299; CI = 0.223 to 0.372; *p* < 0.001).

## 4. Discussion

This systematic review and meta-analysis provides a quantitative review of evidence on nurses’ perceptions of KAP related to the prevention of pressure injuries in hospital settings internationally. This systematic review and meta-analysis provides a focused approach for answering a primary research question by statistically integrating evidence from ten studies. The findings add to the literature on the contribution of nurses’ perceptions regarding their KAP related to the prevention of pressure injury. Nurses’ understanding of pressure ulcers and their prevention plays a vital role in delivering effective patient care. Furthermore, a significant body of research underscores the essential role that nurses’ understanding plays in influencing their attitudes and practices concerning the implementation of pressure injury prevention guidelines [27].

The correlation analysis conducted in this systematic review and meta-analysis revealed a significant positive relationship between knowledge and practice, implying that nurses possessing a greater understanding of pressure ulcer prevention are more inclined to engage in appropriate and favorable attitudes in implementing preventive measures during clinical care. Smith and Waugh [52] reported a similar finding, demonstrating a positive link between nurses’ knowledge and their positive application of pressure ulcer prevention strategies. This alignment in the results emphasizes the need for continuous education and training for nurses to bolster their expertise in this domain. The findings of Murugiah et al. [53] also align with this observation. Their research indicated a deficiency in knowledge among the participants. They posited that an insufficient understanding of pressure ulcer prevention can contribute to a high incidence of pressure ulcers occurring within hospital settings. In addition, a previous study by Zhang et al. [28] reported a positive relationship among nurses’ KAP in preventing pressure injuries, indicating that an increase in the level of knowledge is correlated with favorable attitudes and improved practices.

Nurses’ attitudes toward pressure ulcer prevention can considerably affect their readiness to adopt preventive behaviors. This systematic review and meta-analysis found a significant positive correlation between knowledge and practice toward pressure injury prevention, as indicated in 3 out of the 10 studies included in the final review. This result indicates that nurses with a good understanding are more likely to demonstrate favorable attitudes and practices regarding the implementation of preventive measures for pressure injuries. This finding also aligns with previous research conducted in Australia, which found that nurses with sufficient knowledge also maintained positive attitudes toward the prevention of pressure injuries [54]. A comparable result was observed in a Turkish study, which revealed that intensive care unit nurses possessing adequate knowledge exhibited a positive attitude toward the prevention of pressure injuries [55]. However, two studies included in this systematic review and meta-analysis indicated negative correlations between knowledge and attitudes regarding pressure injury prevention, indicating similar results found in a recent study conducted among nurses in Western Australia [17]. This suggests that as nurses’ knowledge about preventing pressure injuries increases, their level of attitudes diminishes [30,48]. Factors that may contribute to this result include insufficient knowledge regarding pressure injury prevention, heavy nursing workloads, inadequate nursing staff, inadequacy of suitable guidelines for pressure injury prevention, and a scarcity of resources for pressure-relieving equipment, all of which can hinder nurses’ practices in preventing pressure injuries [17,30,48]. In addition, a previous Ethiopian study revealed that nurses who do not possess sufficient knowledge are significantly less likely to exhibit favorable attitudes toward the implementation of effective measures for preventing pressure injuries compared to their well-trained counterparts, thus highlighting the vital role of KAP in providing proficient nursing care [29].

This systematic review and meta-analysis observed a positive correlation between attitude and practice, indicating that nurses with more favorable attitudes toward pressure ulcer prevention are more likely to implement preventive strategies in their clinical routines. There is limited evidence about the relationship between nurses’ attitudes and their compliance with pressure ulcer prevention protocols. However, the result contradicts a previous cross-sectional study carried out in Ethiopia, which revealed that although nurses demonstrated a good understanding and positive attitudes toward pressure injury prevention, their actual practical application did not meet acceptable standards [21]. This inconsistency is supported by findings from several investigations in nursing homes in the United Kingdom [19] and in an emergency hospital in Egypt [20], implying that the effective application of best practices may be hindered by various factors, such as a lack of training, limited resources, and unfavorable attitudes, even though nurses possess adequate knowledge. This inconsistency underscores the need for additional research to explore the intricate dynamics between KAP regarding pressure ulcer prevention among nursing professionals. Ultimately, the actual implementation of pressure ulcer prevention strategies by nurses is a crucial component of patient care. The positive correlation between attitude and practice detected in this systematic review and meta-analysis suggests that nurses with more positive attitudes are more likely to convert those attitudes into effective preventive actions in their clinical practice.

This systematic review and meta-analysis has certain limitations. The findings are applicable solely to registered nurses working in hospital environments. Nevertheless, although the inclusion criteria may have been somewhat stringent, the sample comprised a diverse array of registered nurses from multiple countries, thus showing potential relevance to a broader demographic. In this study, the authors of the 10 studies included in the analysis were not contacted for further information primarily because of concerns regarding time constraints and potential costs. Finally, an important point to note is that this review was limited to studies published in English. Thus, relevant research published in other languages might have been excluded. Finally, while 10 studies were analyzed related to the relationship between knowledge and attitudes, only 3 studies revealed a relationship between knowledge and practice and a relationship between attitudes and practice toward pressure injury prevention.

## 5. Conclusions

This systematic review and meta-analysis identified the significant positive and negative relationships of KAP related to pressure ulcer prevention among nursing professionals. However, when these results are compared with findings from other research, notable discrepancies in connections between these variables emerge. This variation highlights the necessity of additional studies to clarify the intricate relationships among KAP factors and their combined influence on pressure injury prevention in hospital environments.

By improving the knowledge base of registered nurses and promoting a favorable attitude toward pressure ulcer prevention, healthcare organizations have the potential to enhance the already commendable levels of practice and prevention noted in this review. Nursing managers and policymakers ought to develop thorough, evidence-informed educational programs tailored to particular clinical environments to improve nurses’ understanding of pressure injury treatment and prevention.

## Figures and Tables

**Figure 1 healthcare-13-01220-f001:**
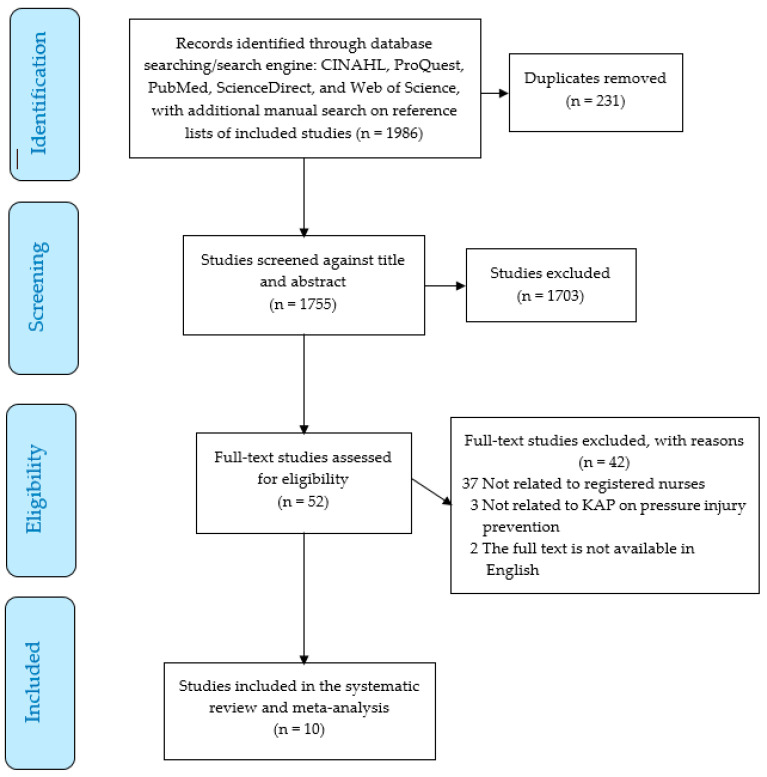
PRISMA flow chart of the systematic review and meta-analysis.

**Figure 2 healthcare-13-01220-f002:**
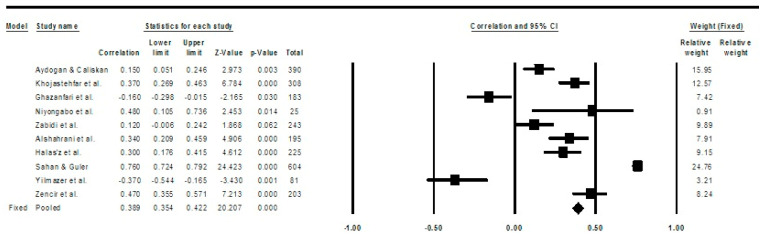
Forest plot for meta-analysis on the correlation between knowledge and attitudes regarding pressure injury prevention with a fixed-effects model. Note: CI = confidence interval, significant if *p* < 0.05 [30,40,41,42,43,44,45,46,47,48].

**Figure 3 healthcare-13-01220-f003:**
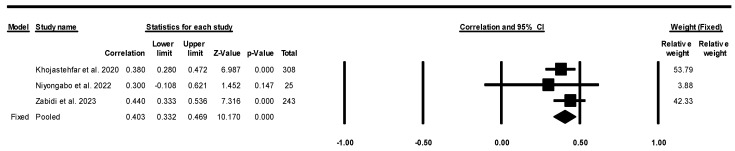
Forest plot of meta-analysis on the correlation between knowledge and practice regarding pressure injury prevention with a fixed-effects model. Note: CI = confidence interval, significant if *p* < 0.05 [43,44,46].

**Figure 4 healthcare-13-01220-f004:**
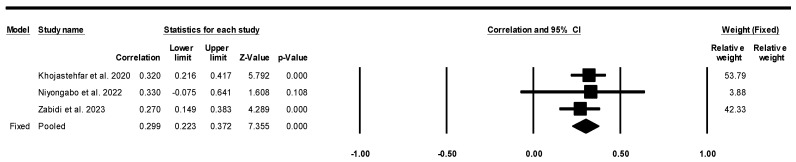
Forest plot for meta-analysis on the correlation between attitude and practice regarding pressure injury prevention with a fixed-effects model. Note: CI = confidence interval, significant if *p* < 0.05 [43,44,46].

**Table 1 healthcare-13-01220-t001:** Quality appraisal according to JBI checklists for analytical cross-sectional studies.

Author(s) and Year of Publication	1. Were the Criteria for Inclusion in the Sample Clearly Defined?	2. Were the Study Objects and the Setting Described in Detail?	3. Was the Exposure Measured in a Valid and Reliable Way?	4. Were Objective, Standard Criteria Used for Measurement of the Condition?	5. Were Confounding Factors Identified?	6. Were Strategies to Deal with Confounding Factors Stated?	7. Were the Outcomes Measured in a Valid and Reliable Way?	8. Was an Appropriate Statistical Analysis Used?	Total Score	Share of Answers YES	Quality Assessment
Zencir et al. [40]	Yes	Yes	No	Yes	Yes	Yes	Yes	Yes	7/8	87.5%	Excellent
Şahan & Güler [41]	Yes	Yes	Yes	Yes	Yes	Yes	Yes	Yes	8/8	100%	Excellent
Alshahrani et al. [42]	Yes	Yes	Yes	Yes	Yes	Yes	Yes	Yes	8/8	100%	Excellent
Zabidi et al. [43]	Yes	No	Yes	Yes	Yes	Yes	Yes	Yes	7/8	87.5%	Excellent
Ghazanfari et al. [30]	Yes	Yes	Yes	Yes	Yes	Yes	Yes	Yes	8/8	100%	Excellent
Niyongabo et al. [44]	Yes	No	Yes	Yes	Yes	Yes	Yes	Yes	7/8	87.5%	Excellent
Halász et al. [45]	Yes	Yes	Yes	Yes	Yes	Yes	Yes	Yes	8/8	100%	Excellent
Khojastehfar et al. [46]	Yes	Yes	Yes	Yes	Yes	Yes	Yes	Yes	8/8	100%	Excellent
Aydoğan & Çalışkan [47]	Yes	Yes	Yes	Yes	Yes	Yes	Yes	Yes	8/8	100%	Excellent
Yilmazer et al. [48]	Yes	Yes	Yes	Yes	Yes	Yes	Yes	Yes	8/8	100%	Excellent

**Scoring:** 1 (Yes), 0 (No), 0 (Unclear), NA (Not Applicable). **Quality assessment:** excellent (>75.1%), some limitations (50.1–75%), and several limitations (≤50%).

**Table 2 healthcare-13-01220-t002:** Study characteristics.

Author/s, Year of Publication, and Country of Origin	Study Aim/Objective	Method	Instrument Used	Nurse (n)	Key Findings	Limitations	Quality Appraisal Score (JBI)
Zencir et al. [40] Turkey	To evaluate the knowledge and attitudes of surgical nurses regarding the prevention of pressure injuries	Descriptive and cross-sectional in nature	Pressure Ulcer Prevention Knowledge Assessment Instrument (PUPKAI-T) Attitude Toward Pressure Ulcer Prevention Instrument (APuP)	203	- Understanding of surgical nurses regarding pressure injury prevention was inadequate. - Nurses demonstrated a positive attitude toward preventing pressure injuries, but this optimistic perspective did not attain a satisfactory standard. - Extent of knowledge held by the nurses was a crucial factor influencing their favorable attitudes.	The findings of the research are primarily derived from the self-reports of the nursing staff. Additionally, the investigation was limited to a single institution, which restricts the generalizability of the results to the broader context of Turkey. Lastly, a pilot study was not performed to assess the feasibility and applicability of the research methodology prior to comprehensive data collection.	7/8
Şahan & Güler [41] Turkey	To determine nurses’ knowledge levels and attitudes regarding pressure injury (PI) and to reveal the relationship between these two variables	Quantitative, descriptive study	Pressure Ulcer Knowledge Assessment Tool (PUKAT) 2.0 Attitude toward Pressure Ulcer Prevention (APuP) The study did not investigate the practice of preventing pressure injury	604	- Average total score for nurses using the PUKAT 2.0 was 9.40. - Average total score for nurses on the APuP instrument was 32.39. - Positive correlation was found between the total knowledge score and the total attitude score (*p*-value = 0.007). - Demographic factors such as gender, educational background, years of experience, and unit of employment were significant predictors of both PUKAT 2.0 scores and APuP scores.	This study presents findings that are limited to a single country, which restricts the ability to generalize the results to the broader nursing population. A comparison of the current results with existing literature reveals discrepancies in the overall and subscale scores related to knowledge and attitudes, as well as in the comparison of these two domains and the factors influencing them. These differences may stem from variations in sample sizes and the specific scales employed to assess knowledge and attitudes regarding pressure injury prevention, particularly considering that the PUKAT scale has undergone revisions. Utilizing a unified scale to evaluate nurses’ knowledge and attitudes toward PI could yield more precise and widely generalizable findings.	8/8
Alshahrani et al. [42] Saudi Arabia	To explore nurses’ knowledge and attitudes toward pressure injury prevention before and after implementing an educational intervention	Pre- and post-intervention study design	Pressure Ulcer Knowledge Assessment Tool (PUKAT 2.0) Attitude toward Pressure Ulcer Prevention (APuP) The practice of pressure injury prevention was not part of this study	Pre-inter-ven-tion phase = 190 Post-inter-ven-tion phase = 195	- Average pre-intervention scores of nurses’ knowledge and attitudes toward pressure injury prevention were documented at 43.22% and 74.77%, respectively. - Following the educational intervention, the scores demonstrated a notable increase, achieving 51.22% for knowledge and 79.02% for attitudes. - A greater comprehension of pressure injury prevention was positively associated with more favorable attitudes toward preventive measures. - Variables such as age, clinical nursing experience, and exposure to intensive care environments were recognized as significant factors influencing knowledge regarding pressure injury prevention. - Having a bachelor’s degree or higher was linked to enhanced knowledge and attitudes regarding pressure injury prevention.	One notable limitation of the study is the lack of distinct participant codes, which hinders the evaluation of knowledge and attitude changes on an individual basis. This constraint restricts the capacity to analyze the correlation between the intervention and the observed changes. Future research should incorporate strategies to effectively monitor participants throughout the various evaluation stages. Furthermore, the absence of a clear criterion for determining adequate knowledge within the PUKAT2.0 instrument constrained the comprehensiveness of the assessment.	8/8
Zabidi et al. [43] Saudi Arabia	To assess nurses’ knowledge, attitude and practices (KAP) toward pressure injury prevention	Cross-sectional descriptive design	Nurses’ Knowledge of Pressure Ulcer Prevention & Management Questionnaire Nurses’ Attitude of Pressure Ulcer Prevention Questionnaire Nurses’ Practice of Pressure Ulcer Prevention Questionnaire	243	- Only one-third exhibited a positive attitude toward pressure injury preventive strategies. - Majority showed acceptable levels of practice and adherence with protocols concerning the prevention and management of pressure ulcers.	The limitation is that, although the study identified particular correlations among nurses regarding their KAP related to pressure injury prevention, comparisons with other studies demonstrate some inconsistencies in these associations. This variation emphasizes the necessity for additional research to clarify the intricate relationships in KAP, as well as their combined influence on pressure injury prevention within clinical environments.	7/8
Ghazanfari et al. [30] Iran	To investigate the knowledge, attitude, and practice (KAP) of Iranian ICU nurses related to pressure injury prevention	Quantitative, cross-sectional study	Pieker Pressure Ulcer Knowledge Test (PPUKT) Attitude toward Pressure Ulcer Prevention (APUP) Practice of nurses related to pressure injury prevention	183	- Average scores indicated KAP of ICU nurses related to pressure injury prevention were measured at 70.57, 52.82, and 22.44, respectively. - Positive correlation was found between the nurses’ attitudes and their practices. - Negative correlation was noted between their knowledge and attitudes. - Positive correlation was identified between the length of the nurses’ experience in the ICU and their knowledge of pressure injury prevention.	The study reported that self-reporting poses a response bias regarding the perceptions of nurses regarding their KAP related to pressure injury prevention.	8/8
Niyongabo et al. [44] Burundi, Africa	To assess nurses’ knowledge, attitude, and practice regarding pressure injury prevention and treatment	Cross-sectional study design	Researchers developed a self-report questionnaire on the knowledge, attitudes, and practices of pressure injury prevention based on two published clinical practice guidelines: the Pan Pacific Guideline for the Prevention and Management of Pressure Injury (2012) published by the Australian Wound Management Association and the Association for the Advancement of Wound Care (AAWC) Venous	25	- Scores reflecting the knowledge and practical skills of nurses were significantly deficient (below 50%). - Attitude scores surpassed 65% across the five items intended to assess attitudes. - Negative correlation was found between the knowledge and attitude scores of the nurses. - Education was inversely associated with the nurses’ knowledge and practice scores. - High attitude score did not align with an increased practice score, which may be explained by the low knowledge scores recorded (under 50% on knowledge items).	This study exclusively examined the perspectives of nurses and nursing assistants, thereby omitting other healthcare professionals from consideration. Additionally, the investigation was carried out in a single public hospital with a capacity of 187 beds, despite the presence of five public hospitals in the city of Bujumbura in Burundi.	7/8
Halász et al. [45] Slovakia	To determine the knowledge and attitudes of nurses toward pressure injury prevention and to find relationships and differences among selected variables	Quantitative, exploratory cross-sectional design	Pressure Ulcer Knowledge Assessment Tool (PUKAT) Attitude toward Pressure Ulcer Prevention (APuP) The study did not explore the practice of pressure injury prevention	225	- Nurses demonstrated insufficient knowledge (45.5%). - Attitudes were 67.9%. - Significant positive correlation was found between their knowledge and attitudes. - Level of education and the specific department of work were linked to significant variations in the knowledge levels of the nurses.	The limitations of this study were associated with the limited availability of information regarding the incidence of pressure ulcers, which is notably sparse. The responsibility for data collection and reporting lies with the healthcare system, which may be reluctant to disclose information that could reflect poorly on the quality of care provided. Furthermore, the findings of this study indicate a significant lack of knowledge and appropriate attitudes concerning pressure injury prevention. Consequently, this raises concerns about the adequacy of incidence recording and whether the data accurately represent the true situation.	8/8
Khojastehfar et al. [46] Iran	To investigate knowledge, attitude, and practice of nurses on pressure injury prevention and their related factors	Quantitative, cross-sectional study with correlational design	Pieper Pressure Ulcer Knowledge Test (PUKT) Attitude toward Pressure Ulcers (APuP) Practice of Pressure Ulcer Prevention	308	- Average scores indicating the knowledge, attitudes, and practices of nurses concerning pressure injury prevention were measured at 63.47 ± 10.31, 39.10 ± 40.22, and 32.03 ± 6.17, respectively. - With each additional year of experience in the ICU, nurses’ knowledge increased by 0.051 units. - Female nurses exhibited greater levels of knowledge and attitude than their male colleagues. - Nurses’ attitudes improved by 0.43 units for every extra hour worked weekly. - Nurses in the general ICU demonstrated higher attitude and practice scores compared to other nursing personnel, with increases of 2.144 and 2.574 units, respectively. - Nurses’ practice improved by 0.162 units for each additional year of age.	In this study, the PUKT, which was established in 1995, was utilized to evaluate nurses’ understanding of pressure ulcers. Since that period, advancements in research related to pressure ulcers have been made, potentially rendering the instrument less representative of the most current knowledge in the field. Therefore, undertaking a similar investigation is advisable, employing newly developed assessment tools.	8/8
Aydoğan & Çalışkan [47] Turkey	To identify the level of knowledge, attitudes, and perceptions of the barriers encountered in pressure injury prevention () among intensive care unit (ICU) nurses	Quantitative, cross-sectional, correlational design	Pressure Ulcer Prevention Knowledge Assessment Instrument (PUPKAI-T) Attitude Toward Pressure Ulcer Prevention Instrument (APuP) This study did not assess the practice of pressure ulcer prevention; instead, it explored the barriers to pressure ulcer prevention	390	- Average knowledge score was found to be 11.54, whereas the average attitude score was 42.96. - Main obstacles to effective pressure injury prevention were identified as insufficient staffing levels (85.6%) and a deficiency in pressure redistribution materials and equipment (82.6%). - Regression analysis of the attitude scores revealed that various factors affected nurses’ attitudes toward pressure injury prevention: self-efficacy in performing pressure ulcer risk assessments (B = 0.28), a motivation to gain additional knowledge regarding pressure injury prevention (B = −0.15), gender (B = −0.15), and the knowledge score itself (B = 0.14).	Due to the high workload in their Intensive Care Units (ICUs), several hospitals in Ankara, Turkey, indicated their inability to participate in the study, resulting in a limitation to six hospitals. Additionally, nurses who were on maternity or sick leave were excluded from the study. Furthermore, neonatal ICUs were not included in the study, as the data collection instruments were deemed inappropriate for that setting.	8/8
Yilmazer et al. [48] Turkey	To assess the knowledge and attitudes of nurses toward pressure injury prevention in ICUs	Cross-sectional study design	Tool for pressure ulcer information Attitude toward pressure ulcer tool (APuP)	81	- A total of 48.1% of nurses employed in ICUs exhibited insufficient understanding of pressure injury prevention. - A total of 21% of nurses attained attitude scores of 75% or above. - Negative correlation was found between the knowledge levels and attitudes of nurses regarding pressure injury prevention (*p* < 0.05).	This study was carried out in the ICUs of a single university hospital; therefore, the findings cannot be generalized to all nursing professionals.	7/8

## Data Availability

The data presented in this study are included in this paper.

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
