# Peer review of "Nurses’ Knowledge, Attitudes, and Practices in Pressure Injury Prevention: A Systematic Review and Meta-Analysis"

_healthcare, 2025, doi:10.3390/healthcare13111220_

Round 1
Reviewer 1 Report
Comments and Suggestions for Authors
Dear authors
Thank you for submitting this interesting review for publication, could you consider the following modifications:
Title: replace toward by in to make the title more concise and clear
Abstract: add appropriate justification indicating the gab for conducting this s. review.
Introduction: it is well structured presenting the current evidence and highlighting the gab in literature, but the transition objective and aim at the end of introduction need to be merged and revised. for example: Given the disparities in existing research, this systematic review and meta-analysis aims to assess the level of knowledge, attitudes, and practices among nurses, and ------.
Methods: add more details on the quality assessment of the included articles
Discussion: you need to link your findings with previous research that not included in your review
suggest longitudinal studies and propose interventions
Thank you
Author Response
Thank you for submitting this interesting review for publication, could you consider the following modifications:
Response: Thank you very much and we considered your suggested modifications, and we revised our paper based on them. We earnestly hope that this will help the editor in favor of accepting our paper for publication in Healthcare.
Title: replace toward by in to make the title more concise and clear
Response: We replaced ‘toward’ by ‘in’ as suggested.
Abstract: add appropriate justification indicating the gab for conducting this s. review.
Response: This has been added in Lines 27-33.
Introduction: it is well structured presenting the current evidence and highlighting the gab in literature, but the transition objective and aim at the end of introduction need to be merged and revised. for example: Given the disparities in existing research, this systematic review and meta-analysis aims to assess the level of knowledge, attitudes, and practices among nurses, and ------.
Response: We revised this part in Lines 179-182.
Methods: add more details on the quality assessment of the included articles
Response: More details have been added in Lines 279-291.
Discussion: you need to link your findings with previous research that not included in your review
suggest longitudinal studies and propose interventions
Response: These have been remarkably addressed in the majority of the Discussion section as suggested. Please refer to Lines 579-638.
Reviewer 2 Report
Comments and Suggestions for Authors
This systematic review and meta-analysis provides valuable insight into the current state of nurses’ knowledge, attitudes, and practices (KAP) toward pressure injury prevention. The article is generally well-structured, methodologically sound, and clearly focused on an important clinical issue with global relevance.
Strengths:
• The topic is timely and clinically significant.
• Use of PRISMA guidelines and JBI appraisal tools enhances methodological quality.
• Data synthesis is thorough, with clear presentation via forest plots and descriptive analysis.
Suggestions for Improvement:
1. Some findings, particularly the negative correlations between knowledge and attitudes in certain studies, should be more deeply discussed to explore possible explanations.
2. Several studies in the review assessed only knowledge and attitude, but not practice. Consider highlighting this as a limitation more clearly.
3. The heterogeneity in tools used (e.g., PUKAT 2.0, APuP, PUPKAI-T) should be discussed in terms of its impact on result comparability.
4. There are occasional issues with language fluency and long sentences that reduce readability. A thorough copy-edit by a native English speaker is recommended.
5. Figure captions should be more descriptive and clearly referenced in the body of the text.
Comments on the Quality of English LanguageSome grammatical inconsistencies, repetition, and lengthy constructions slightly hinder readability. A professional language edit or review by a native English speaker is recommended to enhance fluency and precision.
Author Response
This systematic review and meta-analysis provides valuable insight into the current state of nurses’ knowledge, attitudes, and practices (KAP) toward pressure injury prevention. The article is generally well-structured, methodologically sound, and clearly focused on an important clinical issue with global relevance.
Strengths:
- The topic is timely and clinically significant.
- Use of PRISMA guidelines and JBI appraisal tools enhances methodological quality.
- Data synthesis is thorough, with clear presentation via forest plots and descriptive analysis.
Response: Thank you so much for your valuable review. We hope that your positive feedback will help the editors in their decision toward the acceptance of our paper for publication in Healthcare.
Suggestions for Improvement:
- Some findings, particularly the negative correlations between knowledge and attitudes in certain studies, should be more deeply discussed to explore possible explanations.
Response: This has been addressed in Lines 610-625.
- Several studies in the review assessed only knowledge and attitude, but not practice. Consider highlighting this as a limitation more clearly.
Response: This has been addressed in Lines 655-658.
Response:
- The heterogeneity in tools used (e.g., PUKAT 2.0, APuP, PUPKAI-T) should be discussed in terms of its impact on result comparability.
Response: This has been addressed in Lines 474-479.
- There are occasional issues with language fluency and long sentences that reduce readability. A thorough copy-edit by a native English speaker is recommended.
Response: The revised version of our paper that we resubmitted to the journal has been copy-edited by a native English-speaking editor.
- Figure captions should be more descriptive and clearly referenced in the body of the text.
Response: This has been addressed in Lines 486-531.
Reviewer 3 Report
Comments and Suggestions for Authors
Dear Editor,
Thank you for the invitation to review this manuscript. This is a systematic review and meta-analysis regarding nurses’ knowledge, attitudes, and practices toward injury prevention. The manuscript is well written and easy to read. To enhance the clarity of the manuscript, I have some comments and suggestions for the authors.
Abstract:
- Method: Please clearly state the method of this study in the abstract, that is, this study is a systematic review and meta-analysis.
- Results: Before directly informing 10 studies for meta-analysis, please inform how many studies yielded at the initial search, then how many studies for systematic review, and lastly how many studies were included for meta-analysis.
Introduction:
- Statement on page 2, lines 95–96, saying, “Evidence suggests that nurses who maintain positive attitudes toward safety are associated with a decrease in the occurrence of pressure injuries” need a reference. Please add.
- As mentioned in the title, this study is a systematic review and meta-analysis. I believe the method of the study is indeed systematic review and meta-analysis. However, in some parts of the manuscript, such as in the method of the abstract and at the end of the introduction, the authors mentioned only meta-analysis, as if systematic review was not performed. I suggest the authors mention clearly “systematic review and meta-analysis” in referring to the method of the study.
Methods:
- Why choosing only last five years for the inclusion criteria? Is there any specific reason for this time frame? Please explain.
- On page 4 line 192, it is written that the review examined five databases, one of which is CINAHL. However, in Figure 1 of the PRISMA flowchart, CINAHL is not found in the list of databases of initial search. Was CINAHL used to search the articles? If yes, please add it in the PRISMA flowchart.
- The last paragraph on page 6, including the statement saying p value less than 0.10 from Egger test considered potential bias, needs a reference. Please add the sources for the statements on that paragraph.
Results:
- Before explaining each reviewed study, I suggest the authors start the results section with one paragraph explaining the overall characteristics of the reviewed study, starting from how many studies were included in this review, what the total number of nurse participants is in this study, in which units those nurses worked, in which countries these studies were conducted, what the study designs are, etc.
- Then, followed by explaining what measurement tools were used in the included studies for evaluating nurses’ knowledge, attitude, and practice, and explain further if those validated tools were used in the included studies. For tools using abbreviations, please write the full term for its first use. Please help the readers to understand the content and do not assume that every reader will understand all abbreviations presented in the manuscript.
- On page 8, lines 298–300, it is written that, “demographic factors of gender, educational background, years of experience, and unit of employment were significant predictors of both PUKAT 2.0 scores and APuP scores.” This statement needs further elaboration. Please specify the information, such as what gender (male or female), what level of education (vocational, bachelor, etc.), how many years of work experience, and which unit of employment that significantly predict the scores of pUKAT 2.0 and APuP.
- The same comment for the study by Zabidi et al. (2023) on page 8, lines 319–321. Please elaborate on what gender (female or male nurses), marital status (married or unmarried nurses), educational attainment, and area of practice demonstrated the significant differences for pressure ulcer practice.
- The position of the figure and the related paragraph needs improvement. On page 9, lines 376, the authors presented Figure 2, but there is no information about Figure 2. Then, on page 18, lines 394–395, the authors mentioned the value of CI and correction value in Figure 2, and this paragraph is followed by Figure 3, while Figure 2 is on page 9. This makes it difficult for the readers to read and follow the information presented in the manuscript. I suggest the authors put the table near the paragraph explaining the table.
Discussion
On page 18, line 428, it is written that this meta-analysis integrated evidence from nine studies. Is that nine or ten studies? Please clarify.
Conclusion
The conclusion is clear.
Author Response
Thank you for the invitation to review this manuscript. This is a systematic review and meta-analysis regarding nurses’ knowledge, attitudes, and practices toward injury prevention. The manuscript is well written and easy to read. To enhance the clarity of the manuscript, I have some comments and suggestions for the authors.
Response: We greatly appreciate your support and positive review. We hope that your favorable feedback will assist the editors in their decision for the acceptance of the revised version of our manuscript for publication in the prestigious journal of Healthcare.
Abstract:
- Method: Please clearly state the method of this study in the abstract, that is, this study is a systematic review and meta-analysis.
Response: This has been revised in Lines 35-39.
- Results: Before directly informing 10 studies for meta-analysis, please inform how many studies yielded at the initial search, then how many studies for systematic review, and lastly how many studies were included for meta-analysis.
Response: These have been addressed in Lines 39-45.
Introduction:
- Statement on page 2, lines 95–96, saying, “Evidence suggests that nurses who maintain positive attitudes toward safety are associated with a decrease in the occurrence of pressure injuries” need a reference. Please add.
Response: This has been added in Line 109.
- As mentioned in the title, this study is a systematic review and meta-analysis. I believe the method of the study is indeed systematic review and meta-analysis. However, in some parts of the manuscript, such as in the method of the abstract and at the end of the introduction, the authors mentioned only meta-analysis, as if systematic review was not performed. I suggest the authors mention clearly “systematic review and meta-analysis” in referring to the method of the study.
Response: This has been observed throughout the text in the revised version of the paper.
Methods:
- Why choosing only last five years for the inclusion criteria? Is there any specific reason for this time frame? Please explain.
Response: This has been explained in Lines 204-207.
- On page 4 line 192, it is written that the review examined five databases, one of which is CINAHL. However, in Figure 1 of the PRISMA flowchart, CINAHL is not found in the list of databases of initial search. Was CINAHL used to search the articles? If yes, please add it in the PRISMA flowchart.
Response: This has been addressed in Figure 1 in Line 217, and in the Methods section in Line 261.
- The last paragraph on page 6, including the statement saying p value less than 0.10 from Egger test considered potential bias, needs a reference. Please add the sources for the statements on that paragraph.
Response: The sources have been added in Lines 311-312.
Results:
- Before explaining each reviewed study, I suggest the authors start the results section with one paragraph explaining the overall characteristics of the reviewed study, starting from how many studies were included in this review, what the total number of nurse participants is in this study, in which units those nurses worked, in which countries these studies were conducted, what the study designs are, etc.
Response: This has been added in Lines 315-332.
- Then, followed by explaining what measurement tools were used in the included studies for evaluating nurses’ knowledge, attitude, and practice, and explain further if those validated tools were used in the included studies. For tools using abbreviations, please write the full term for its first use. Please help the readers to understand the content and do not assume that every reader will understand all abbreviations presented in the manuscript.
Response: These have been addressed in Lines 333-364.
- On page 8, lines 298–300, it is written that, “demographic factors of gender, educational background, years of experience, and unit of employment were significant predictors of both PUKAT 2.0 scores and APuP scores.” This statement needs further elaboration. Please specify the information, such as what gender (male or female), what level of education (vocational, bachelor, etc.), how many years of work experience, and which unit of employment that significantly predict the scores of pUKAT 2.0 and APuP.
Response: These have been addressed in Lines 377-384.
- The same comment for the study by Zabidi et al. (2023) on page 8, lines 319–321. Please elaborate on what gender (female or male nurses), marital status (married or unmarried nurses), educational attainment, and area of practice demonstrated the significant differences for pressure ulcer practice.
Response: These have been addressed in Line 403-413.
- The position of the figure and the related paragraph needs improvement. On page 9, lines 376, the authors presented Figure 2, but there is no information about Figure 2. Then, on page 18, lines 394–395, the authors mentioned the value of CI and correction value in Figure 2, and this paragraph is followed by Figure 3, while Figure 2 is on page 9. This makes it difficult for the readers to read and follow the information presented in the manuscript. I suggest the authors put the table near the paragraph explaining the table.
Response: These have been addressed in Line 471-531.
Discussion
On page 18, line 428, it is written that this meta-analysis integrated evidence from nine studies. Is that nine or ten studies? Please clarify.
RESPONSE: This is a typo error and has been corrected as ‘ten’ studies. Thank you for highlighting this.
Round 2
Reviewer 3 Report
Comments and Suggestions for Authors
Dear Editor,
Thank you for the invitation to review the revised version of the manuscript. I appreciate the authors’ efforts to address all comments in the previous version. The manuscript is now better and easier to read. For this revised version, I just have some minor suggestions for the authors below.
1. The time frame of studies reported in this review is not consistent in several parts of the manuscript. On abstract page 1, line 35, and on page 8, line 296, it is written that evidence searched was published between 2019 and 2024. However, on page 4, lines 184 and 197, it is written that reviewed articles were published from 2019 to 2025. Which one is the correct one, 2024 or 2025? Please clarify.
2. Information on page 4, lines 190–193, and on page 5, lines 210–213, about exclusion criteria is redundant. Please remove one and avoid presenting repetitive information.
3. Information starting on page 8, line 329, to page 10, line 415, is the same word-for-word as what is written in Table 2 on pages 11–17. Again, please avoid presenting redundant information and sentences. I suggest the authors paraphrase the sentences in the key findings column on Table 2 and make them simpler. Key findings can be written with bullets with short sentences.
Author Response
Thank you for the invitation to review the revised version of the manuscript. I appreciate the authors’ efforts to address all comments in the previous version. The manuscript is now better and easier to read. For this revised version, I just have some minor suggestions for the authors below.
RESPONSE: We thank you so much for your continued support with much appreciation.
- The time frame of studies reported in this review is not consistent in several parts of the manuscript. On abstract page 1, line 35, and on page 8, line 296, it is written that evidence searched was published between 2019 and 2024. However, on page 4, lines 184 and 197, it is written that reviewed articles were published from 2019 to 2025. Which one is the correct one, 2024 or 2025? Please clarify.
RESPONSE: Thank you for noticing this and it has been corrected to 2025.
- Information on page 4, lines 190–193, and on page 5, lines 210–213, about exclusion criteria is redundant. Please remove one and avoid presenting repetitive information.
RESPONSE: Repetitive information about the exclusion criteria has been deleted. Thank you.
- Information starting on page 8, line 329, to page 10, line 415, is the same word-for-word as what is written in Table 2 on pages 11–17. Again, please avoid presenting redundant information and sentences. I suggest the authors paraphrase the sentences in the key findings column on Table 2 and make them simpler. Key findings can be written with bullets with short sentences.
RESPONSE: Thank you again and we paraphrased the key findings in Table with bullets with short sentences.